# Comparison of Chemical and Mechanical Surface Treatments on Metallic Precision Spheres for Using as Optical Reference Artifacts

**DOI:** 10.3390/ma15113741

**Published:** 2022-05-24

**Authors:** Víctor Meana, Eduardo Cuesta, Braulio J. Álvarez, Sara Giganto, Susana Martínez-Pellitero

**Affiliations:** 1Department of Construction and Manufacturing Engineering, University of Oviedo, Campus de Gijón, 33203 Gijón, Spain; ecuesta@uniovi.es (E.C.); braulio@uniovi.es (B.J.Á.); 2Area of Manufacturing Engineering, Universidad de León, Campus de Vegazana, 24071 León, Spain; sgigf@unileon.es (S.G.); susana.martinez@unileon.es (S.M.-P.)

**Keywords:** laser scanning, reference spheres, chemical attack (etching), sandblasting

## Abstract

The improvement of industrial manufacturing processes requires measurement procedures and part inspection tasks to be faster and faster while remaining effective. In this sense, the capabilities of noncontact measuring systems are of great help, not only because of the great amount of data they provide but also for the ease of the integration of these systems as well as their automation, minimising the impact on the industry. This work presents a comparative study on the influence of two surface treatments performed on low-cost, high-precision metallic spheres on the suitability of these spheres to be used as artefacts for the calibration of optical sensors, specifically laser triangulation sensors. The first surface treatment is sandblasting (a mechanical process), whose effect has been studied and presented in previous work. The second treatment focused on in this paper is acid etching (a chemical process). The comparison has been performed by evaluating the same metrological characteristics on two identical groups of spheres of similar type (diameter and accuracy), each of which was subjected to a different treatment. It was necessary to obtain the reference values of the metrological parameters with high accuracy, which involved measuring the spheres with a coordinate measuring machine (CMM) by contact probing. Likewise, spheres were scanned by a laser triangulation sensor mounted on the same CMM. The results derived from both the contact and laser measurements and before and after treating the surfaces were used to compare four parameters: point density, sphere diameter, sphere form deviation, and standard deviation of the best-fit sphere to the corresponding point cloud. This research has revealed that acid etching produces better optical qualities on the surfaces than the mirror-like original ones, thus enhancing the laser sensor capturing ability. However, such chemical etching has affected the metrological characteristics of the spheres to a greater extent than that produced by sandblasting. This difference is due to the variability of the chemical etching, caused by the high aggressiveness of the acid, which makes the process very sensitive to the time of exposure to the acid and the orientations of the spheres in the bath.

## 1. Introduction

Noncontact metrology plays an important role in many industrial processes, not only in the field of part inspection but also in the application of reverse engineering, particularly in areas such as the automotive or aerospace. Among the four types of methods that have been developed in the noncontact metrology (optical, ultrasonic, pneumatic, and electrical), the first two have continuously evolved in accuracy and resolution (mainly profilometers and laser triangulation sensors) and are nowadays extremely popular, either in automated inter-process inspection [1], or in-process [2].

In this sense, it is essential to find solutions that minimise the impact on the industry of the introduction of noncontact verification methods with the aim of providing traceability of the corresponding measurement results [3], regardless of the field of the application being manual, semiautomated, or completely automated, as occurs with laser sensors mounted on CMMs [4]. Particularly, the setup and calibration of optical sensors [5] are critical to ensure the traceability of measurement results, and therefore, the availability of reference artefacts that can be used effectively in those phases is essential.

Focusing on the metrological aspects of a laser scanner sensor, several factors affect the accuracy of the measurements: the intrinsic characteristics of the sensor [6], the laser beam orientation, the scanning path [7], and the physical and geometric properties of the part [8,9,10]. Specifically, it is also known that laser scanner measurements are highly affected by the optical properties of the part surfaces [11,12] or even by the ambient lighting [13].

Previous research works have revealed that one of the main weaknesses of these sensors is the impossibility of measuring parts with very bright surfaces with high accuracy. The reflectivity of the surface is the source of important randomness in results since it prevents the optical sensor from capturing a sufficient number of points and, furthermore, generates false or spurious points (not belonging to the surface) in the captured point clouds.

Other works [14,15] make it clear that the addition of antireflective coatings with (removable) powders or (fixed) paints is not a sufficiently accurate solution for reference elements, although these investigations on “anti reflective” coatings suggest a beneficial impact on accuracy. Therefore, the solution proposed in this article is to directly modify the surface condition of the reference element without adding any coating.

Therefore, objects used as reference artefacts for the setup and calibration should possess suitable optical properties while maintaining enough dimensional and geometrical accuracy. Within the wide range of calibration artefact geometries, spheres stand out as they allow for implementing several metrological entities, such as diameters, form deviations, and length dimensions by means of the distance between the centres of two of them. This is the main reason that reference spheres are commonly used in interim checking and performance procedures, which are applied for qualifying probes in a multitude of coordinate measuring machines, as well as being used in the calibration of conventional metrology instruments. In fact, spheres are used in many calibration procedures, even establishing the best capturing parameters and determining the conditions for maximum accuracy in measurements performed by machines where laser sensors are mounted [16].

The current spheres used in metrological applications as reference elements are basically spheres of ceramic materials, such as alumina, ruby or zirconia, among others, as well as mixtures between them (e.g., zirconia and alumina). These spheres are manufactured by sintering from powder, machining of preforms and subsequent polishing. This process achieves precision grades G3, G5 and G10 (according to ISO 3290/DIN 5401 [17]), which means achieving a sphericity < 0.25 µm, and *Ra* < 0.020 µm. This high precision means a very shiny and specular surface. However, this finish is not suitable for optical metrology equipment, due to the high reflectivity that makes it impossible to capture enough points and even more so of high quality.

In this experimentation, a low-cost precision stainless steel balls has been chosen due to they are standard balls in the bearing industry. By default, these balls also have good qualities (G50 or G100, according to ISO 3290/DIN 5401) with sphericity < 2.5 µm and *Ra* < 0.1 µm. However, they still have shiny and reflective surfaces, which are not suitable for optical equipment. The surface finish of these spheres is intended to modify in this work, checking that this modification does not affect the quality and precision (dimensional and geometrical) significantly, which would prevent it from being a metrological reference element. The aim of this work is to check whether the chemical attack process using an acid bath modifies the surface, eliminating the shine without losing the grade of quality. Furthermore, the aim is also to quantify the possible improvement in the quality of the point cloud before the acid bath (original state, with gloss) and after the chemical treatment (without gloss).

In previous experimentation [18], it was proved that sandblasted spheres can be employed as reference artefacts for calibrating noncontact sensors, as their form deviation was in the order of 0.004–0.005 mm. Sandblasting generates minimal changes both in the diameter and in the form deviation. On the other hand, the sandblasted texture allows for improving the point cloud capturing, thus providing a better surface coverage which results in a more accurate reconstruction of the sphere.

Precisely, this work is intended to analyse and compare the influence of two different surface treatments, the sandblasting (mechanical) and etching, by immersion in acid (chemical), onto the surface of low-cost precision spheres made of stainless steel. The first objective is to prove that the etching treatment is able to reduce the reflectivity of the sphere surface without significantly altering the geometrical characteristics of the sphere in order to use it in the setup and calibration procedures of optical sensors and other noncontact metrological and reverse engineering equipment. It must be noted that both finishing processes studied here will change the form error and also the dimensions of the original ones. However, the key for a sphere to become “standard” is overall a low form error and low standard deviation to the best fitting sphere. The diameter value is not as relevant for their use as a reference element since, once calibrated, this size value will be known with high accuracy.

To achieve this goal, the experimentation performed in this work encompasses a series of tasks, from the manufacturing of the spheres to the evaluation of such spheres after etching treatment. The equipment that has been used to validate the experimentation is a laser triangulation sensor mounted on the ram of a coordinate measuring machine (CMM). However, higher accurate measurement was also needed to obtain the reference values for the subsequent comparison between the two surface treatments, so a contact probe mounted on the same CMM was also employed. Then, for each sphere and each state (pre and post treatment), two measurements were performed, first by contact probing and second by laser scanning.

Therefore, our study is intended to determine the level of influence of the chemical treatment applied to the spheres for obtaining a less reflective surface, more suitable for capturing point clouds by laser scanning. The collected data will allow for establishing the feasibility and degree of suitability of the treatment.

On the other hand, a second objective is to compare the data resulting from both surface treatments (mechanical and chemical) in order to establish the best conditions of the parameters to analyse. The parameters that have been studied in this work are the number of points in the capture point cloud, the sphere diameter, the sphere form deviation, and the standard deviation of the point cloud regarding the best fit reconstructed sphere. Finally, this research is also aimed at quantifying the improvement in the quality of the point clouds captured by laser scanning after reducing the reflectivity of the surface.

It is important to highlight that the experimentation involved replicating the procedure carried out for the sets of spheres that had been treated mechanically by sandblasting [18], that is, repeating the same steps but applied to a new set of spheres subjected to chemical treatment. The results collected have been used for the comparative analysis of both finishing processes.

## 2. Materials and Methods

Three plates have been designed and manufactured to serve as supports for a sufficient number of identical spheres (10 units per plate) to statistically evaluate the influence of the surface treatments (Figure 1a). Based on the designs of the standard samples and after defining the criteria to fulfil in the measurements, both in contact and in noncontact types, all the parameters, elements, and procedures are established and registered.

Each plate platform was manufactured in stainless steel AISI 316L and previously sandblasted to avoid reflections. This platform supports 10 spheres of identical size, identical material, AISI 316L with quality grade G100, sphericity below 2.5 µm, and arithmetic mean roughness *Ra* < 0.1 µm. The system chosen to join the spheres to the plates was the threaded joints. In each precision sphere, a hole was drilled and then tapped. Special care was taken in the design of the fixture (a vise with hemispherical jaws) for clamping the spheres in order to prevent marking or deforming the sphere surface. The precision spheres were drilled using two hemispherical jaws manufactured ad hoc for each diameter in such a way that no permanent marking or deformation occurred. Subsequently, each sphere hole was threaded, which allowed the 10 spheres to be mounted (screwed) on each plate. All the spheres used in the experimentation are precision made of AISI 316 stainless steel, grade G100, with a sphericity of less than 2.5 µm and a roughness *Ra* < 0.1 µm.

In total, considering the mechanical and chemical treatments, the experimentation has been carried out on over 60 spheres of different sizes (20 units of Ø10 mm size, 20 units of Ø20 mm, and 20 units of Ø25 mm). Each sphere has been measured 10 times by contact probing and 10 times by laser scanning before and after the respective surface treatment, which accounts for 40 measurement routines. Therefore, a collection of 2400 sets of measurement results, clustered in three groups of 800 measurements per sphere size, has been collected for the statistical analysis of the defined parameters.

The contact probing measurement was performed in a CMM model DEA Global Image 091508 (Figure 2a) with a Maximum Permissible Error in Length Indication MPE_L_ = 2.2 + 3·L/1000 (µm, L in mm) and a Maximum Permissible Error in Repeatability MPE_E_ = 2.2 µm calibrated according to ISO 10360-2. This CMM mounts a Renishaw SP25 sensor on an indexable head Renishaw PH10MQ. The probe tip selected was a 1.5 mm diameter ball of synthetic ruby. The choice of this tip, conditioned by the stem size and the ball diameter, is considered suitable for measuring the three sizes of spheres without probe changes, thus ensuring a greater uniformity in the results of different batches and trying to minimise the possible influence of external factors in the measurements.

On the other hand, the equipment used for noncontact measurement was a laser triangulation sensor HP-L-10.6 from Hexagon Metrology (Figure 2b) also mounted on the same CMM. It is a sensor with a FOV (Field Of View) of 170 ± 30 mm, with three magnification options, which leads to three line widths (24/60/124 mm) and two options for the distance between points at each magnification setting, according to the frequency or line rate, being 53 Hz the maximum value. The control software used for programming and executing the measurement routines was PC-DMIS 2018 R2 for both contact and laser sensors. Measurements took place in the same laboratory at a controlled temperature of 20 ± 5 °C as indicated in the ISO 1:2002 [19].

The experimentation was developed in five phases corresponding to the following activities:Manufacturing and the assembly of the plates with spheres. Three plates, with sets of 10 spheres of equal size per plate (Ø10 mm, Ø18 mm, and Ø25 mm), were manufactured. The arrangement of the spheres, as well as the identical assembly on each plate, make the handle easier and favours the univocal identification of each sphere at its location (Figure 1b);Contact and noncontact measurements of the original spheres. Firstly, all the spheres on the plates were measured by contact probing at the CMM in order to obtain reference values, both dimensional and geometrical, with great accuracy. Additionally, the spheres were digitised by laser scanning using the aforementioned sensor;Surface treatment of the artefacts. Similar to the previous experimentation [18], where 3 sets of spheres (10 spheres of size Ø10 mm, 10 of Ø18 mm, and 10 of Ø25 mm) had been subjected to a sandblasting operation with aluminium oxide particles, now new sets of spheres (10/10/10) are chemically etched by immersion in a bath consisting of hydrochloric acid (HCl) at 35% and nitric acid (HNO_3_) at 65% (molar ratio 3:1) for 8 min. The surface finishing of the spheres is modified, thus obtaining sets with less brightness and different texture;Contact and noncontact measurements of all treated sets. Both the sandblasted sets and the etched ones were measured again after the corresponding treatment by contact probing in the CMM first and second, by digitising with a laser triangulation sensor;Analysis of results. Finally, the measurement results obtained before and after treating the sets of spheres by sanding and chemical etching were compared. In this comparison, the values of the measured dimensions and deviations regarding geometrical tolerances were first obtained by contact measurement and subsequently by analysis of the point clouds acquired by noncontact measurement.

The first device employed for measuring the sets of spheres was the CMM equipped with a contact probe. The objective of this contact measurement was to determine the reference values of those dimensions and geometric tolerance deviations selected for the comparative analysis. The minimum density for the contact probing was set to 0.2 points/mm^2^ so that 40 points were probed on each of the Ø10 mm spheres, 100 points on the Ø18 mm spheres, and 200 points on the Ø25 mm spheres, measuring only the upper hemispheres. These sets of points were distributed differently on the surface of the spheres according to the sphere size, leading to 5, 7, and 10 rows (meridians) for the location of the probing points. On the other hand, the noncontact measurement was carried out by a laser sensor mounted on the ram of the CMM. This sensor can adjust the light intensity point to point (10 times/point), thus providing an excellent optical range. The point density was set to 16.8 points/mm with a line width of 123 mm to acquire the best quality and reliable point clouds. The sensor was located in five different orientations (Figure 3) to cover a high portion of the spherical surface, even acquiring points below the horizontal equator of the sphere: A0B0, A45B0, A45B180, A45B90, and A45B-90 according to the terminology of the Renishaw^®^ PH10MQ motorised probe head.

Although the programming time for the automatic scanning in PC-DMIS^®^ was about 2 h, the program execution time was less than 2 min for each plate, including all scanning orientations.

Before obtaining the measurement data from the point clouds scanned with the laser triangulation sensor, a previous study was carried out aimed at determining the best type of filter for “cleaning” the point clouds. From previous research works [20], it is known that the best filter for subsequent reconstruction of an entity from the filtered point cloud is the Sigma or Standard Deviation (σ) filter. This filter discards or removes points located at a distance (from the reconstructed entity) larger than *x* times the standard deviation of the point cloud (usually *x* is 2 or 3, leading to 2σ or 3σ filters).

The study for the best type of filter and its value was performed with the aid of the Geomagic^®^ Control X 2020 (3D Systems, Rock Hill, SC, USA) software. The methodology followed in this study involved:Reconstruction of the entity (“best fit” sphere) without applying any filter;Determination of the standard deviation of the point cloud with regard to the “best fit” sphere;Calculation of the value of the filter by multiplying the standard deviation and the selected factor (2, 3, …) for analysis;Reconstruction of the entity (“best fit” sphere) after applying the filter with the value calculated in the previous step.

This methodology was applied because the value of the standard deviation filter defined by default in the software was not able to provide the expected results, as it was demonstrated in previous research [20]. For a better understanding of the proposed methodology, it is summarised in Figure 4.

For both contact and noncontact measurements, different fixturing systems were arranged for clamping the sets of plates and spheres, selecting for each case the optimal solution that maximises the accessibility of the probe (Figure 5a) or the visibility of the laser triangulation sensor (Figure 5b) when measuring the spherical components.

The sandblasting treatment was carried out using aluminium oxide WFA F100 as the abrasive (Figure 6a). In the present research, spheres were subjected to a chemical etching by immersion in aqua regia solution, to which 3 g of iron chloride (FeCl_3_) were added for dyeing. The different sphere sets were immersed in a glass recipient filled with the solution so that it only came into contact with the precision spheres. After 8 min of immersion, each sphere set was extracted and rapidly immersed in another recipient filled with deionised water in order to stop the chemical reaction and to remove the acid from the sphere’s surface. Finally, spheres were air-dried, leading to the result that can be observed in Figure 6b.

Once the processes for modifying the surface finish of the spheres with aqua regia were completed, new measurements were performed on the sets of spheres, again by contact probing in the CMM (Figure 7) and by noncontact digitising with the laser triangulation sensor (Figure 8).

In a similar manner to the original measurements of the spheres, before any surface treatment, in the procedure of noncontact measurement of the treated spheres, these were captured using 5 different orientations of the laser sensors that allow the generation of the point clouds shown in Figure 9a. This number of orientations was sufficient to capture, at least, the upper hemisphere of each sphere. The processing of the point clouds, carried out with the aid of Geomagic^®^ Control X 2020 (3D Systems, Rock Hill, SC, USA), involved, in the first place, the removal of those points not pertaining to the spheres (points of the base plate and the auxiliary fixture devices) so that the point clouds to be addressed are only those relating to the upper hemispheres.

In the second place, points located below the horizontal equator of each sphere were erased (Figure 9b). In the last step, a 2σ filter is applied to remove spurious points, which are clearly located far away from the sphere surface and whose inclusion in the analysis would distort all the performed measurements.

## 3. Results

Once all the measurements were performed, either by contact or without contact, over the original spheres or the chemically etched spheres, the obtained results were thoroughly analysed. Three parameters were compared: the diameter of the spheres, the form deviation or sphericity, and the standard deviation of the sphere reconstructed from a point cloud. On the one hand, a comparison was made between the original and the etched spheres. On the other hand, a later comparison also considered the results of sandblasted spheres obtained in previous experimentation.

### 3.1. Pre- and Post-Acid Treatment

Table 1 and Table 2 collect the average measurement results obtained for the different sizes of spheres according to their different states (original or pre-acid treatment and post-acid treatment) and corresponding to the different measurement technologies (contact and noncontact). Table 1 is focused on the results of diameter measurements, whereas Table 2 includes the form deviations. It should be noted that original spheres were scanned using parameters of low sensibility in the laser sensor settings because when high sensibility values are set, the sensor is not capable of acquiring a suitable number of points for reconstructing the spherical geometry with enough reliability [18].

Regarding the diameter measurement results (Table 1), the differences between contact and noncontact measured values have been substantially smaller in the case of post-acid-treatment spheres than in the original state. While initially, the differences are in the order of one-tenth of a millimetre, they decrease to the half in the case of the chemically etched spheres, that is, until the order of five-hundredths of a millimetre for Ø10 mm and Ø25 mm spheres, and even reaching 0.030 mm for Ø10 mm spheres (all measurements performed under high gain mode). In any case, nonhomogeneous values are observed for the parameters measured on the Ø10 mm spheres as a difference with regard to the measured values for the Ø18 mm and Ø25 mm spheres.

Qualitatively, similar results were observed for the measurements of the form deviation of the spheres (Table 2). Differences between contact measurement and noncontact measurement results were again higher before etching the spheres. In this case, the differences in the etched spheres, setting high gain parameters in the laser sensor, reached an average value of 71 µm in the three sphere sizes, whereas before etching, the form deviation attained values in the order of magnitude of tenths of a millimetre, even up to 0.5 mm for Ø10 mm spheres.

Regarding the surface finish of the treated spheres, using a TESA Rugosurf10^®^ (TESA, Renens, Switzerland) roughness tester, it was observed that average values for *Ra* were above 1 µm, that is, at least ten times the average roughness *Ra* that was measured on the original spheres (*Ra* < 0.1 µm). Figure 10 shows, as an example, the roughness measured values in one of the Ø25 mm spheres, in their original state and after mechanical and chemical treatments.

### 3.2. Acid Bath versus Sandblasting Treatment

Table 3 collects the diameters measured by contact probing in the CMM of the sets of spheres after each of the two surface treatments, as well as the differences between those values and the measured diameters of the original spheres. It is evident that sandblasting causes slight increments in the diameter values (always lower than 3 µm), whereas the effect of chemical etching generates a reduction in the diameter that can be quantified by an average value of 61 µm for all the sizes of spheres.

In the same sense, Table 4 gathers the values of the form deviation of the spheres measured by contact probing. In this case, the differences in the sandblasted spheres with regard to the original state are almost imperceptible (in the range of 1 or 2 µm), while a maximum difference of 25 µm is reached in the Ø18 mm spheres after being immersed in aqua regia.

Focusing on the analysis of the point cloud acquired by the laser triangulation sensor, it is confirmed that the quantity and quality of the points are improved with regard to the original spheres because of the removal of reflections caused by the brightness of the surfaces. The parameter that best characterises the quality of the point cloud is the standard deviation [5]. In fact, this parameter is considered a good substitute for the form deviation in the case of evaluating the quality of the point cloud when it is fitted to a sphere.

Figure 11 presents the values of the standard deviation of the point clouds corresponding to each of the 10 spheres included in the six sets analysed in this experimentation (three sandblasted and three acid-treated). Continuous lines represent the values corresponding to the three sizes of the spheres in their original state prior to the treatments. Dashed lines show the values of the treated spheres, distinguishing by colours those corresponding to sandblasted spheres and those corresponding to chemically etched spheres. The graph shows clearly how values of sandblasted spheres are uniform and centred around 0.012 mm while values of etched spheres are more dispersed (from 0.0177 mm for the 9th Ø25 mm sphere to 0.0428 mm for the 7th Ø18 mm sphere).

On the other hand, Table 5 includes the number of points captured before and after the two surface treatments, showing the improvement of these point clouds with respect to the first point clouds. A larger increment has been detected in the case of the smaller spheres (Ø10 mm), which can be explained by the relation between the density of the laser line and the sphere diameter. Nevertheless, in the three sizes, the increase in the number of captured points is greater in the case of the etched spheres than in the case of sandblasted spheres.

The improvement in the density and coverage of the point clouds was evident, as the coverage in the original spheres was very poor. In fact, in some cases (Ø10 mm spheres), not all the spheres could be reconstructed correctly. As a consequence, the comparisons between the pre- and post-sandblasted spheres and between the pre- and post-acid-treated spheres were only possible in all the sphere sizes when a high gain mode was employed (low sensitivity of the laser sensor).

Another improvement is related to the closeness of agreement between dimensional results derived from laser measurements and those derived from contact measurements (reference). This improvement can be evaluated by means of the percentual ratio between the parameter measured with the laser sensor and the corresponding parameter measured by contact probing in the CMM. For instance, an improvement of 100% in any parameter would imply that laser measurements provide an equal result to contact measurements.

Applying this evaluation to the case of the sandblasted spheres (Figure 12), it can be seen that the larger improvement occurs for the larger spheres (Ø18 mm and Ø25 mm), where all the improvement ratios are very high (>60%). Actually, the measured diameters are very close to the reference values. Regarding the form deviation, the average improvement ratio for the three sphere sizes was 63.80%. Finally, for the standard deviation of the point clouds, the average improvement ratio of the three sphere sizes reached 59.21%.

A similar evaluation was carried out for the chemically etched spheres, and the results also show that improvements were achieved in the values obtained after performing this surface treatment in aqua regia. The improvements are substantial, although nonhomogeneous, for all the sphere sizes (Figure 13). As a difference with regard to the sandblasted spheres, which also showed better results after the treatment, it can be observed that the improvement ratios are substantially lower, reaching values between 30% and 40% in the cases of the form deviation and standard deviation for the Ø18 mm and Ø25 mm spheres, and values lower than 75% for the diameter for those same sphere sizes.

## 4. Conclusions

The improvement of the quality of point clouds acquired by noncontact measurement systems and the metrological parameters extracted from them has been evaluated by comparing the measurement results of metallic spheres before and after being subjected to two surface treatments: sandblasting and chemical etching by immersion in aqua regia. To ensure the repeatability of the experiments, the study was carried out employing a high number of spheres clustered according to three different sizes on different plates. Moreover, several repetitions of measurements were also performed, both in the contact measurements (by contact probing in a CMM) that were taken as reference and in the noncontact measurements (by scanning with a laser triangulation sensor mounted on the same CMM).

The first conclusion that can be extracted is that the treatment in aqua regia causes reductions in the diameters of the three sphere sizes that can be quantified in the range of 0.08 mm, as evaluated by contact measurements (see “CMM contact” columns in Table 1). However, when spheres are scanned with the laser sensor, results show that the chemical treatment causes, on the one hand, a reduction in the diameter measurement of the smaller spheres (Ø10 mm), but on the other hand, an increase in the diameter measurement of the medium (Ø18 mm) and large size (Ø25 mm) spheres. This different tendency could be attributed to the influence of surface roughness on the contact measurement, which is greater for the smaller spheres.

The comparison of contact and noncontact measurement results obtained after the surface treatment revealed differences in diameter oscillating around an average value of 0.043 mm (−0.0478 with Ø10 mm, −0.0299 with Ø18 mm, −0.0527 with Ø25 mm). It is also observed that, once the spheres have been treated, the differences in the form error values obtained by contact and noncontact oscillate around 0.071 mm.

Secondly, the study confirms the improvements in the quality and quantity of the points belonging to the point clouds scanned by the laser sensor for all the sphere sizes after the chemical treatments, with ratios that even outperform the improvements achieved using sandblasting. The improvements are noticeably more significant in the case of the smaller spheres (76.78% in the chemically treated spheres and 61.72% in the sandblasted spheres).

Regarding the standard deviation of point clouds scanned before and after the two surface treatments, the analysis reveals a reduction in this parameter after treating the spheres. This reduction is lower in the case of the chemical treatment than in the sandblasting treatment. Likewise, the range of values of this parameter for each position of the spheres is noticeably larger in the case of the chemically treated spheres, with results oscillating between 0.0174 mm and 0.0428 mm, while the values obtained from the sandblasted spheres reflect a greater homogeneity (between 0.0093 mm and 0.0126 mm).

Thirdly, it can be stated that the mechanical treatment of spheres by means of sandblasting produces a better dimensional agreement between contact and noncontact measurements than the chemical treatment. This result can be checked for the three analysed parameters (diameter, form deviation, and standard deviation of the point cloud), although it is more clearly observed in the case of medium and large spheres (Ø18 mm and Ø25 mm).

This research proves that the chemical treatment is able to minimise the light reflections on the sphere surface, thus enhancing point capture by laser scanning, but it also modifies substantially the analysed metrological parameters, which invalidates this process for obtaining spheres that can be used as reference elements. Therefore, the objective of improving the values obtained by sandblasting, whose validity had been proved in previous work, is not achieved. It is important to note that the immersion in an acid bath is more aggressive as the time during which the stainless steel sphere is immersed increases. In fact, after immersing the spheres for 8 min, a subsequent cleansing was needed, and the final air drying has generated marks on the sphere surface in the form of meridians caused by gravity, which has affected the surface finish (Figure 14).

A complete control of the way to attack, i.e., the exposure-to-acid time and the position of the spheres, and even the drying process, could help achieve better results than those obtained within this research, which undoubtedly can constitute the basis of future work.

## Figures and Tables

**Figure 1 materials-15-03741-f001:**
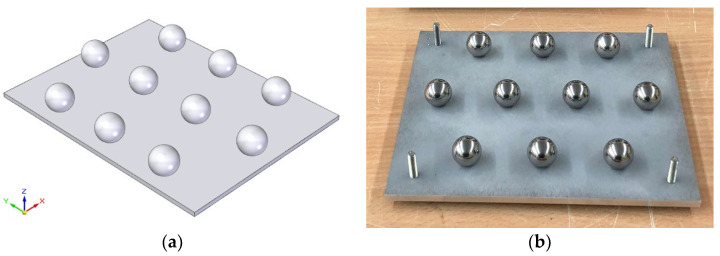
(**a**) Basic design of the spheres plate. (**b**) Plate with Ø18 mm precision spheres before acid etching.

**Figure 2 materials-15-03741-f002:**
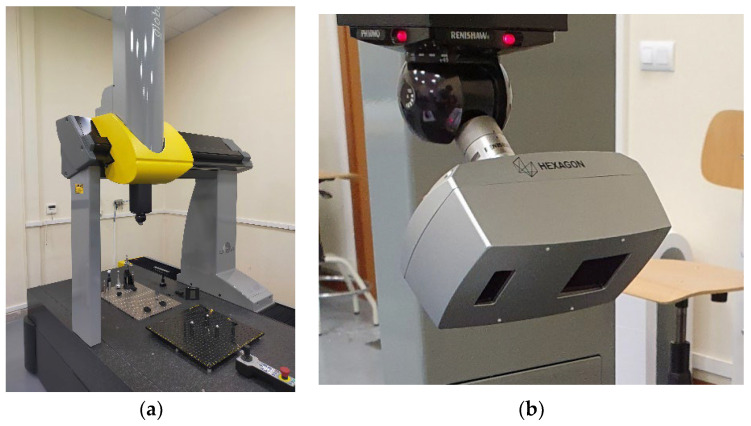
(**a**) coordinate measurement machine; (**b**) laser triangulation sensor.

**Figure 3 materials-15-03741-f003:**
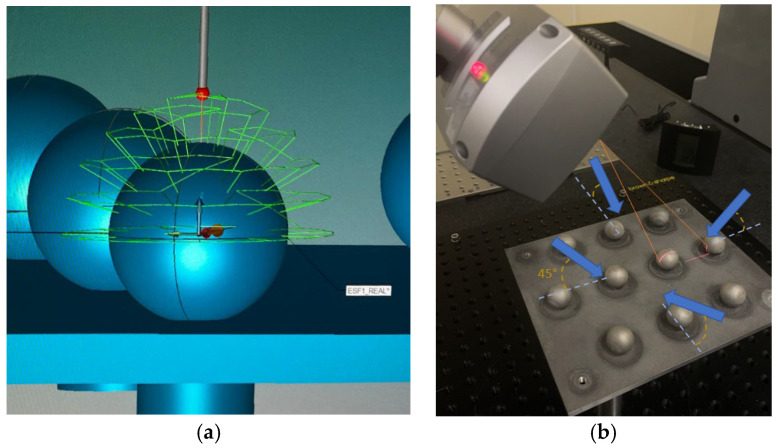
(**a**) Contact measurement (probing strategy for the Ø10 mm sphere). (**b**) Noncontact measurement (sensor scanning in the A45B90 orientation).

**Figure 4 materials-15-03741-f004:**
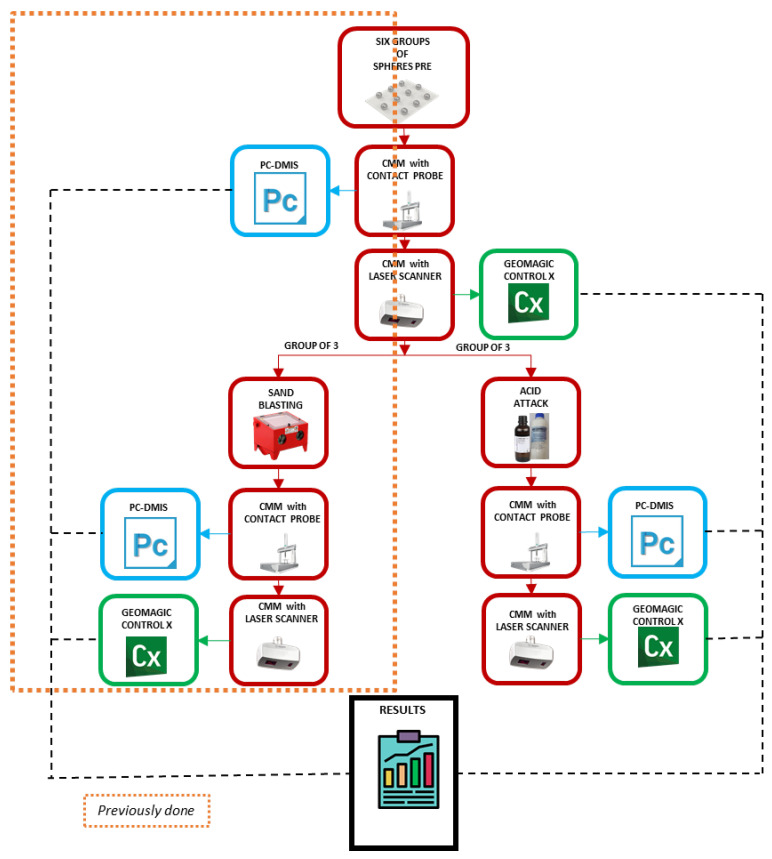
The full methodology of the experimentation.

**Figure 5 materials-15-03741-f005:**
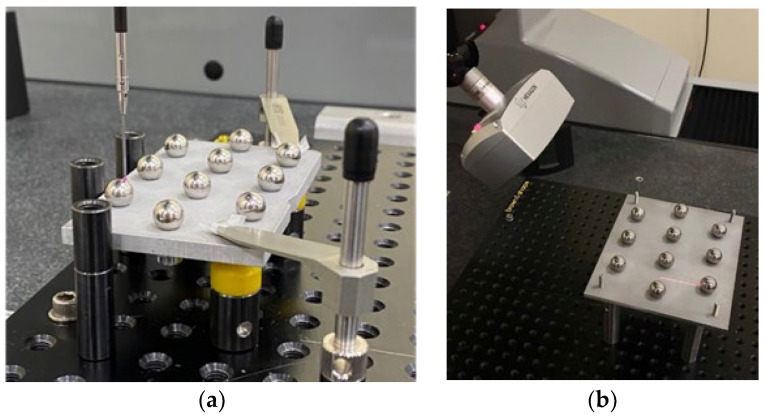
(**a**) Modular fixture for contact measurements. (**b**) Fixture by simple support for noncontact measurements.

**Figure 6 materials-15-03741-f006:**
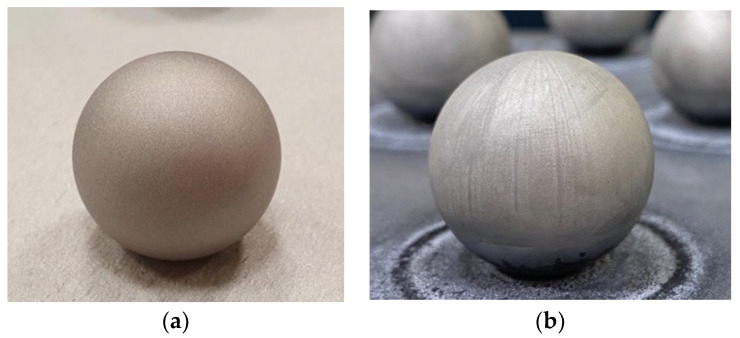
(**a**) Surface finish after sandblasting (previous research). (**b**) Surface finish after chemical treatment with aqua regia.

**Figure 7 materials-15-03741-f007:**
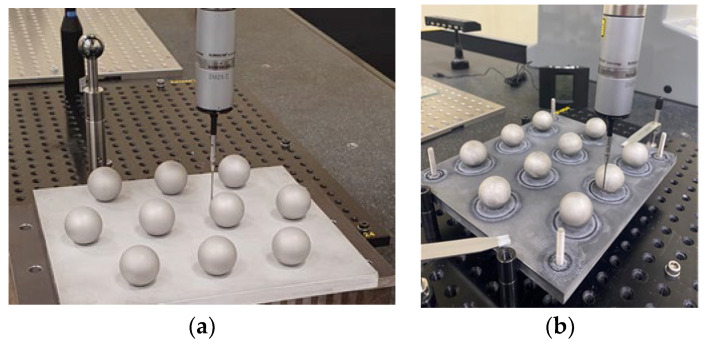
Contact measurements of sets of spheres after each treatment. (**a**) After sandblasting (**b**) After chemical etching.

**Figure 8 materials-15-03741-f008:**
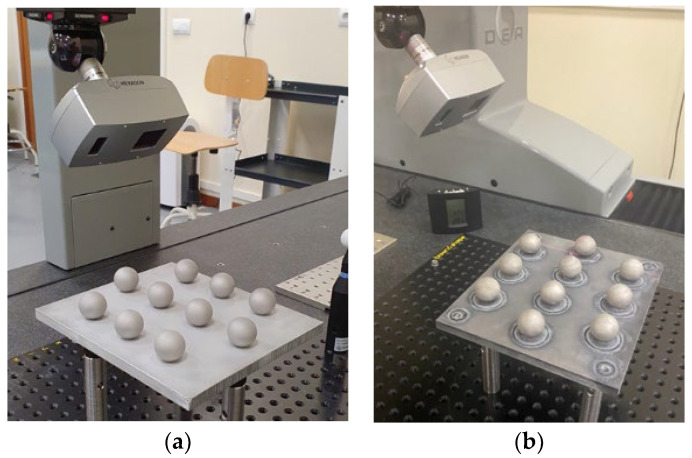
Noncontact measurements of sets of spheres after each treatment. (**a**) After sandblasting (**b**) After chemical etching.

**Figure 9 materials-15-03741-f009:**
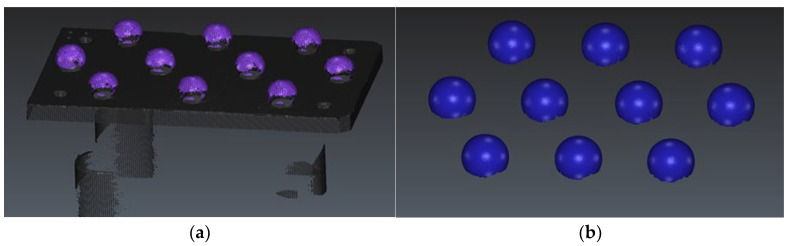
(**a**) Point cloud digitised by the laser sensor Hexagon HP-L-10.6. (**b**) Point clouds corresponding to the different spheres after cleaning and filtering.

**Figure 10 materials-15-03741-f010:**
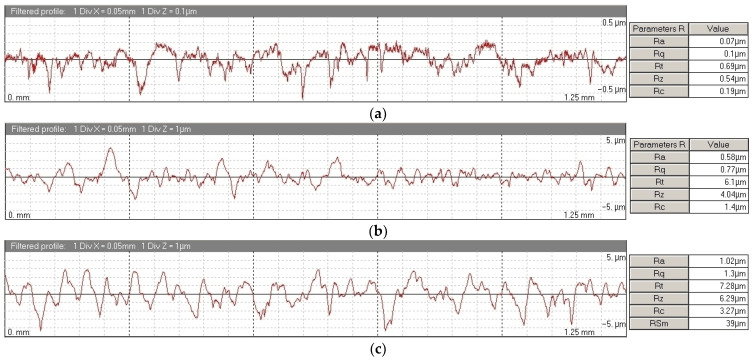
An example of the roughness profile (cutoff length = 0.25 mm) and the measured parameters of a Ø25 mm sphere: (**a**) original, (**b**) post-sandblasting, and (**c**) post-acid treatment.

**Figure 11 materials-15-03741-f011:**
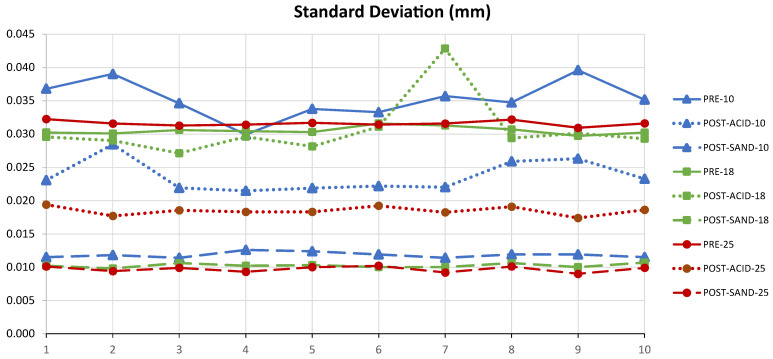
Standard deviation values pre and post sandblasting and acid bath.

**Figure 12 materials-15-03741-f012:**
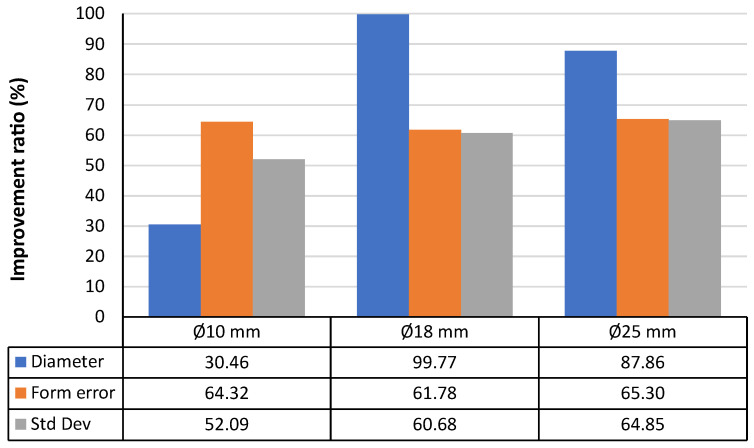
Improvement ratio in the noncontact measurement for the three parameters studied and the three diameters considered after the sandblasting process.

**Figure 13 materials-15-03741-f013:**
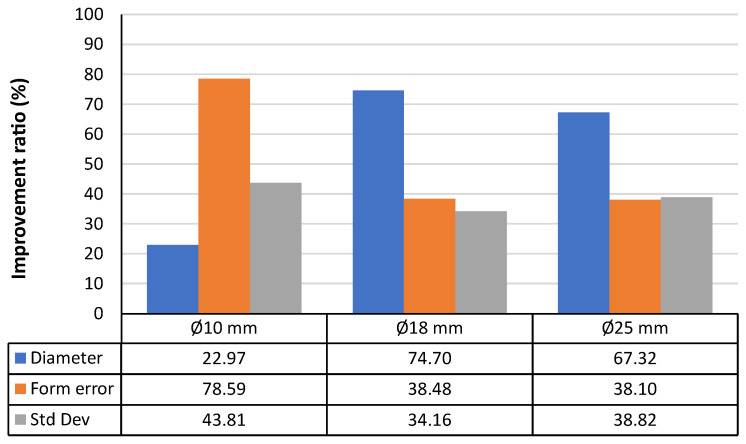
Improvement ratio in the noncontact measurement for the three parameters studied and the three diameters considered after the chemical bath process.

**Figure 14 materials-15-03741-f014:**
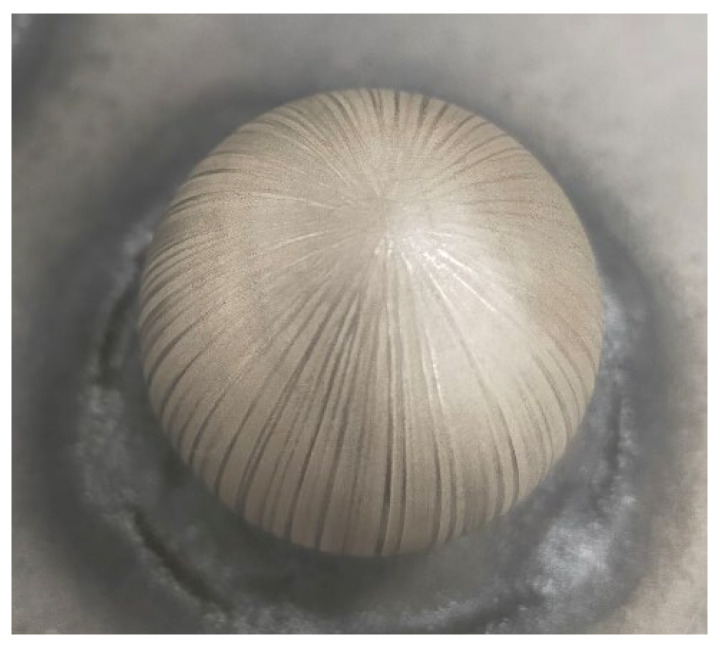
Ø25 mm sphere with surface markings.

**Table 1 materials-15-03741-t001:** Comparison of contact and noncontact measurements (mm) regarding diameter pre- and post-acid treatment.

Sphere Size	Pre-Acid Treatment			Post-Acid Treatment		
	CMM (Contact)	Laser Sensor		CMM (Contact)	Laser Sensor	
	Diameter	Diameter	Difference	Diameter	Diameter	Difference
Ø10 mm	10.0018	9.9629	−0.0389	9.9492	9.9014	−0.0478
Ø18 mm	18.0065	17.8884	−0.1181	17.9554	17.9255	−0.0299
Ø25 mm	25.0080	24.8468	−0.1611	24.9293	24.8767	−0.0527

**Table 2 materials-15-03741-t002:** Comparison of contact and noncontact measurements (mm) regarding form error pre- and post-acid treatment.

	Pre-Acid Treatment			Post-Acid Treatment		
	CMM (Contact)	Laser Sensor		CMM (Contact)	Laser Sensor	
Sphere Size	Form Error	Form Error	Difference	Form Error	Form Error	Difference
Ø10 mm	0.0022	0.4917	0.4895	0.0247	0.1053	0.0805
Ø18 mm	0.0021	0.1446	0.1426	0.0276	0.0890	0.0614
Ø25 mm	0.0049	0.1476	0.1427	0.0190	0.0914	0.0724

**Table 3 materials-15-03741-t003:** Comparison of contact measurement results regarding diameter pre and post treatments.

Sphere Size	Average Diameter [mm]			
	Post-Acid	Post-Sand	Difference (Acid)	Difference (Sandblasting)
Ø10 mm	9.9492	10.0054	−0.0526	0.0028
Ø18 mm	17.9554	18.0004	−0.0511	0.0027
Ø25 mm	24.9293	25.0075	−0.0786	0.0026

**Table 4 materials-15-03741-t004:** Comparison of contact measurement results regarding form deviation pre and post treatments.

Sphere Size	Average Form Deviation [mm]			
	Post-Acid	Post-Sand	Difference (Acid)	Difference (Sandblasting)
Ø10 mm	0.0247	0.0046	0.0225	0.0010
Ø18 mm	0.0276	0.0044	0.0255	0.0021
Ø25 mm	0.0190	0.0047	0.0141	0.0020

**Table 5 materials-15-03741-t005:** Comparison number of points in the point clouds.

Sphere Size	Number of Point Cloud			Point Clouds Number Improve	
	Pre	Post-Acid	Post-Sand	Post-Acid	Post-Sand
Ø10 mm	8699	15,030	14,068	72.78%	61.72%
Ø18 mm	39,303	45,640	42,836	16.12%	8.99%
Ø25 mm	68,971	85,821	79,586	24.43%	15.39%

## Data Availability

Not applicable.

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
