# Peer review of "Comparison of Chemical and Mechanical Surface Treatments on Metallic Precision Spheres for Using as Optical Reference Artifacts"

_materials, 2022, doi:10.3390/ma15113741_

Round 1
Reviewer 1 Report
In this work, a comparative study was presented on the influence of two surface treatments (sandblasting and acid etching) performed on metallic spheres on the suitability of these spheres to be used as artefacts for the calibration of optical sensors, specifically laser triangulation sensors. The comparison has been performed by evaluating the same metrological characteristics on two identical groups of spheres of similar type (diameter and accuracy), each of which was subjected to a different treatment. The paper is significant for the industrial application of non-contact metrology, but the following comments should be addressed.
- The effect of scanning angle during laser Triangulation Sensor should be considered, because the light reflections may be different under various scanning angles.
- The acquisition process and time to point clouds should be added in the part of Materials and methods. Whether or the acquisition times should be 3 times for one spheres.
- The AISI 316L plates were selected as substrate in the manuscript. What is the effect of AISI 316L plate on the light reflection.
Author Response
See attached document

Reviewer 2 Report
The submitted manuscript discusses experiments surrounding a novel chemical etchant procedure aimed at improving laser-based non-contact metrology measurements, specifically the density and accuracy of generated point clouds. Overall the work is an interesting topic and the study seems well thought out and performed. To further improve the manuscript I offer the following suggestions:
- An extra review for English writing style would be appropriate. I understood everything presented with the current writing, but several sentences were a bit clunky and the paper could read more smoothly with some touch-up, although this may just be my stylistic preference.
- Consider replotting/reformatting some of the figures. In particular figure 11 is a bit hard to follow, and would be even more difficult for anyone who prints in black-and-white ink. Perhaps consider breaking it up into 3 subfigures, one for each of the sphere diameters.
- The screen capture approach used in figure 10 is not an appropriate figure, and is mostly illegible in its current form. Replot the data with clear axis and descriptions, or at least format the image so it doesn't show excess information from the software interface.
- I think the manuscript could be strengthened with some extra discussion on the implications and use case for the findings presented. More specifically, I would like to better understand the final use case: would these spheres just be a reference for metrology, or would the various shapes/parts being monitored also have this acid etch applied. Furthermore does the findings of 10mm spheres underestimating size as compared to the other sizes have implication for which types of parts/geometries it is an appropriate reference shape for? Would a complex part with fine and large features need more than one reference sizes? I don't think this discussion needs to be extensive, but some follow up on the real use case for these findings would improve the paper.
Author Response
See attached document

Reviewer 3 Report
This manuscript fulfilled the target to compare chemical and mechanical surface treatments on metallic reference spheres for using as optical reference artifacts from the scope of the work. More details are needed for publish.
- I think data of pre-sand treatment is used to calculated the difference before and after sandblasting. So the data should be listed out in a table like pre-acid treatment.
- In Figure 12 and 13, after both sandblasting and acid treatment, 10mm ball shows a higher form error and a lower diameter error, while 18 and 25 mm ball are opposite. Can you explain? Do you have a similar figure for pre-treatment measurement?
Author Response
See attached document
